# Long-term viable chimeric nephrons generated from progenitor cells are a reliable model in cisplatin-induced toxicity

Kenji Matsui [1], Shuichiro Yamanaka [1,5✉], Sandy Chen[1], Naoto Matsumoto[1], Keita Morimoto[1], Yoshitaka Kinoshita [1,2], Yuka Inage[1,3], Yatsumu Saito[1], Tsuyoshi Takamura[1], Toshinari Fujimoto[1], Susumu Tajiri[1], Kei Matsumoto[1], Eiji Kobayashi[4] & Takashi Yokoo [1,5✉]

Kidney organoids have shown promise as evaluation tools, but their in vitro maturity remains limited. Transplantation into adult mice has aided in maturation; however, their lack of urinary tract connection limits long-term viability. Thus, long-term viable generated nephrons have not been demonstrated. In this study, we present an approachable method in which mouse and rat renal progenitor cells are injected into the developing kidneys of neonatal mice, resulting in the generation of chimeric nephrons integrated with the host urinary tracts. These chimeric nephrons exhibit similar maturation to the host nephrons, long-term viability with excretion and reabsorption functions, and cisplatin-induced renal injury in both acute and chronic phases, as confirmed by single-cell RNA-sequencing. Additionally, induced human nephron progenitor cells differentiate into nephrons within the neonatal kidneys. Collectively, neonatal injection represents a promising approach for in vivo nephron generation, with potential applications in kidney regeneration, drug screening, and pathological analysis.

[1] Division of Nephrology and Hypertension, Department of Internal Medicine, The Jikei University School of Medicine, Tokyo 105-8461, Japan. [2] Department of Urology, Graduate School of Medicine, The University of Tokyo, Tokyo 113-8654, Japan. [3] Department of Pediatrics, The Jikei University School of Medicine, Tokyo 105-8461, Japan. [4] Department of Kidney Regenerative Medicine, The Jikei University School of Medicine, Tokyo 105-8461, Japan. [5] These authors jointly supervised this work: Shuichiro Yamanaka, Takashi Yokoo. ✉email: shu.yamanaka@jikei.ac.jp; tyokoo@jikei.ac.jp

The development of human-induced pluripotent cells (hiPSCs)[1] has facilitated organoid studies that emulate the developmental process, which are anticipated as a valuable evaluation tool[2]. Although in vitro renal organoids lag behind the maturity of native fetal kidneys[3,4], their maturation can be stimulated by transplantation under the renal capsule, benefiting from the host blood supply[5–7]. However, the lack of connection to the urinary tract limits their long-term viability[8]. Establishing a connection between exogenous nephrons and the host urinary tract would extend their viability and enhance their clinical utility.

Two main strategies have been employed to generate chimeric nephrons. The first involves injecting stem cells into blastocysts, which requires microinjection techniques and transfer of injected embryos into pseudopregnant foster mice[9]. The second strategy involves introducing exogenous nephron progenitor cells (NPCs) into developing fetal kidneys[10–14]. During nephrogenesis, NPCs form cap mesenchyme and differentiate into podocytes to distal tubules while interacting with ureteric buds and stromal progenitor cells[15,16]. Our previous work described the exo utero method, which involved injecting exogenous mouse renal progenitor cells (RPCs) into the retroperitoneal cavities of fetal mice through the amniotic membrane. However, with this method, precise injection into the fetal kidneys is challenging, and it requires foster mothers to raise the hosts after cesarean section[11].

To establish a more practical and efficient model for generating chimeric nephrons, we have focused on neonatal mice, which retain NPCs until approximately postnatal day 3 (P3)[17,18]. During this critical period, exogenous NPCs would be integrated into the host cap mesenchyme. Previous studies have demonstrated that neonatal mouse kidneys can serve as scaffolds for developing nephrons. However, these studies have only assessed the exogenous nephrons for a limited duration, up to 2 weeks[19–22]. Therefore, further investigation was needed to assess their long-term viability, functionality, and response to nephrotoxic insults. In this study, we have developed the neonatal niche injection (NNI) method, which proved to be highly approachable for generating chimeric nephrons. Using this method, chimeric nephrons derived from exogenous mouse (allogeneic) and rat (xenogeneic) progenitor cells are generated, exhibiting maturation levels comparable to host nephrons. These chimeric nephrons with a connection to the host urinary tract remain viable over an extended period, and are valuable for evaluating acute and chronic nephrotoxicity. Furthermore, hiPSC-derived NPCs also differentiate into nephrons within neonatal mouse kidneys with a technical modification, although further confirmation of chimerism is required.

## Results

**Development of the NNI method.** Six2-positive NPCs were identified in the outermost layer of neonatal mouse kidneys (Fig. 1a)[17,18]. Donor RPCs were obtained by dissociating fetal mouse kidneys of E14.5 C57BL/6-Tg (CAG-EGFP) mice (EGFP mice). Kidneys of neonatal (P0.5 or P1.5) C57BL/6NCrSlc mice (B6 mice) were exposed through a back incision, and RPCs were injected under the renal capsule with a 34 G Hamilton syringe under a fluorescent stereomicroscope (Fig. 1b, c and Supplementary Movie 1). The operation time per host was $10.0 \pm 0.5$ min ($n = 23$, mean ± SEM). Following the injection, the host neonates were nurtured by their biological mothers (Fig. 1d). The survival rates at 2 weeks post-injection were 43% in P0.5 neonates ($n = 44$) and 85% in P1.5 neonates ($n = 39$), respectively (Table 1). Importantly, the injection procedure did not negatively affect the growth of the host neonates (Fig. 1e).

**Successful integration and differentiation of donor cells in the neonatal nephrogenic niche.** On day 2 (P3.5), the host cap mesenchyme remained in the outer layer (Fig. 2a). The EGFP-positive donor NPCs were integrated into the cap mesenchyme (Fig. 2a–c). Two weeks later, the presence of EGFP-positive cells was confirmed in all hosts, indicating a significant enhancement in injection accuracy compared to the exo utero method (Table 1)[11]. The donor cells demonstrated localized engraftment (Fig. 2d and Table 2). Immunostaining revealed chimeric glomeruli (Fig. 2e, f) and both proximal and distal tubules (Fig. 2g, h). The chimeric glomeruli exhibited recruitment of vascular endothelial cells derived from the EGFP-negative host (Fig. 2f). Chimeric collecting ducts and mesangial cells were also generated, reflecting that fetal RPCs contain not only NPCs but also stromal progenitor cells and ureteric buds (Supplementary Fig. 1a–c)[11].

**Comparable maturation of the donor nephrons to the host.** The maturation of chimeric nephrons 2 weeks post-injection was evaluated using single-cell RNA-sequencing (scRNA-seq), comparing donor-derived cells defined by *EGFP* RNA expression and host cells not expressing *EGFP*. First, to assign individual cells to a particular clustering, four samples were analyzed separately and integrated: short-term chimeras without cisplatin (CIS) administration (2 weeks, CIS (−)), acute CIS-induced kidney injury model of short-term chimeras (2 weeks, CIS (+)), long-term chimeras without CIS (2 months, CIS (−)), and chronic repeated-dose CIS-induced kidney injury model of long-term chimeras (2 months, CIS (+)) (Fig. 3a). EGFP-positive regions of the host kidneys were trimmed off (Supplementary Fig. 2a, b) and enzymatically dissociated into a single-cell suspension. From a total of 24,682 cells, those with similar transcriptional profiles were clustered and labeled based on the expression of the previously described marker genes (Fig. 3b and Supplementary Fig. 2c)[23–25]. These four groups neatly overlapped in the clusters (Fig. 3c). The total number of *EGFP*-expressing donor cells was 423 (1.7%) and they were distributed in nephron epithelial cells including proximal tubule cells (PTCs, PT-S1 to S3, Fig. 3b, d). Gene expression patterns of nephrons in the 2 weeks, CIS (−) sample showed high correlation between host and donor cells, indicating comparable maturation (Fig. 3e). Furthermore, in PTCs, expression levels of the transporters involved in CIS uptake and excretion (Supplementary Fig. 2d)[26] were not significantly different between host and donor (Fig. 3f). The expression of CTR1 and MATE1 in chimeric tubules was also confirmed by immunostaining (Fig. 3g).

To assess whether in vitro culture can also support the maturation of PTCs, the RNA expression levels of those transporter genes were compared in reverse-transcription quantitative polymerase chain reaction (RT-qPCR) among E14.5 fetal kidneys, P9.5 mature kidneys, and renal spheroids prepared from E14.5 B6 mouse kidneys and cultured on the air–liquid interface for 7 or 12 days (Supplementary Fig. 3a–f). The expressions of *Slc22a1*, *Slc22a2*, *Slc47a1*, and *Abcc2* in the in vitro-cultured spheroids were significantly lower than in the mature kidneys, although higher than in the fetal kidneys. Moreover, prolonged culture periods did not improve maturation (Supplementary Fig. 3g). These findings suggest that in vitro culture may not adequately support tubular maturation compared to chimeric nephrons.

**Cisplatin-induced acute kidney injury model using chimeric nephrons.** A CIS-induced acute kidney injury model was created using these sufficiently mature chimeric nephrons (Fig. 4a). CIS was intraperitoneally administered to the hosts, resulting in a dose-dependent increase in the expression level of kidney injury molecule-1 (Kim1) in PTCs derived from both host and donor cells (Fig. 4b–d). To further evaluate the nephrotoxicity, two samples were compared in scRNA-seq: 2 weeks, CIS (−) and 2 weeks, CIS (+) (Fig. 3a). First, the extent of CIS-induced

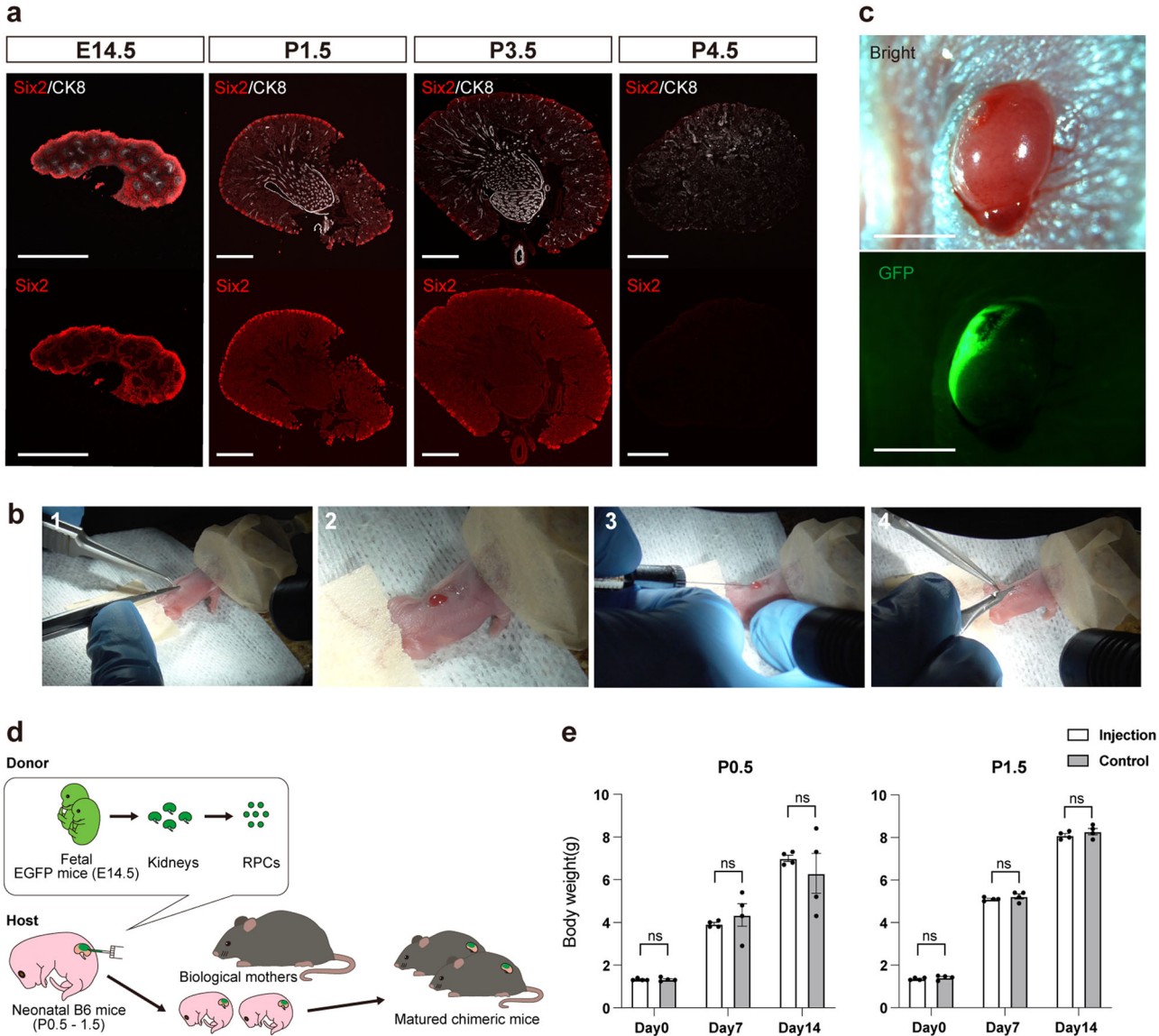

**Fig. 1 Neonatal niche injection (NNI) method. a** Residual Six2-positive nephron progenitor cells (NPCs) in the kidney of fetal (E14.5) and neonatal (P1.5, P3.5, and P4.5) mice. In neonates, NPCs are located in the outermost layer, whereas the differentiated nephrons are in the medulla. **b** Procedures of the NNI method. (1) Under anesthesia, a vertical incision is made on the left back. (2) Light pressure is applied to expose the left kidney out of the incision. (3) Approximately 1 µL of suspension (1.0 × 10⁶ cells) of renal progenitor cells (RPCs) is injected under the renal capsule with a 34G Hamilton syringe. (4) The incision is sutured with 8-0 nylon. All procedures were performed under the stereomicroscope. **c** Fluorescence stereomicroscopic images of the host neonatal mouse kidney after the injection. GFP-positive exogenous RPCs are confirmed to spread on the surface. **d** A schematic of the NNI method. **e** Bodyweight changes in the host neonates that received the injection on P0 and P1 compared with non-surgical controls. No significant differences were found between groups ($n = 4$ biologically independent samples). Error bars represent the mean ± SEM. Data were analyzed using the two-tailed unpaired t-test. ns, not significant. Scale bars, 500 µm in (**a**) and 2 mm in (**c**). CK8, cytokeratin 8; GFP, green fluorescent protein.

changes in individual gene expression showed a significant correlation between host and donor cells, indicating similar responses (Fig. 4e). In particular, genes highly upregulated or downregulated by CIS were comparable between the host and donor cells (Fig. 4f). The upregulated genes included known acute kidney injury markers such as *Havcr1* (Kim1), *S100a6*, *Clusterin*, *Cdkn1a*, and *Spp1*[27–29]. Conversely, the downregulated genes were mature transporter markers such as *Slc34a1*[27] and genes involved in glycogenesis such as *Pck1*[30], consistent with previous reports (Fig. 4g).

**Long-term viability and function of chimeric nephrons**. As a comparison, EGFP-positive mouse RPC were cultured overnight

to form spheroids and transplanted under the renal capsules of adult female NOD/Shi-scid, IL-2RgKO Jic mice (NOG mice) that are immunocompromised. The spheroids differentiated in 2 weeks but did not integrate into the host nephrons. Their tubules got dilated after 1 month and disrupted after 2 months (Fig. 5a, b). In contrast, neonatal chimeric nephrons remained viable in all cases even after 2 and 4 months of cell injection (Fig. 5c–g). In addition, systemically administered fluorescent-labeled low-molecular-weight dextran was observed around the EGFP-positive podocytes and on the apical surfaces of EGFP-positive PTCs, which suggested the functional filtration and reabsorption capacity of chimeric nephrons (Fig. 5c, d). The gene expression patterns of each nephron remained highly correlated between the host and donor, particularly in PTCs, even after long-term engraftment (Fig. 5h).

**Table 1 Comparisons between neonatal niche injection and blastocyst complementation, as well as fetal injection.**

| | Neonatal niche injection (P1.5) | Blastocyst complementation (E2.5)[9] | Fetal injection (*Exo utero*, E13.5)[11] |
|---|---|---|---|
| Second surgery | Not required | Embryo transfer to pseudopregnant mice | Cesarean section and transfer to foster mothers |
| Host survival rate | 85% (33 of 39)[a] | 43% (34 of 79)[b] $p < 0.0001$[c] | 74% (14 of 19)[d] $p = 0.47$[c] |
| Percentage of donor nephron engraftment in kidneys relative to surviving hosts | 100% (33 of 33) | 71% (24 of 34)[e] $p < 0.0001$[c] | 14% (2 of 14) $p < 0.0001$[c] |
| Percentage of donor nephron engraftment in kidneys relative to all injected hosts | 85% (33 of 39) | 30% (24 of 79) $p < 0.0001$[c] | 11% (2 of 19) $p < 0.0001$[c] |

[a]Hosts that survived 2 weeks after the surgery performed by the skilled experimenter.
[b]Percentage of pups that reached adulthood from embryos derived from intercrosses of *Sall1*[+/−] mice injected with embryonic stem cells.
[c]Comparisons against neonatal niche injection using the Chi-square test.
[d]Percentage of fetuses that survived until cesarean section performed by the skilled experimenter. Note the subsequent risk of being killed by foster mothers.
[e]Percentage of adults derived from embryos injected with embryonic stem cells, exhibiting confirmed chimera formation.

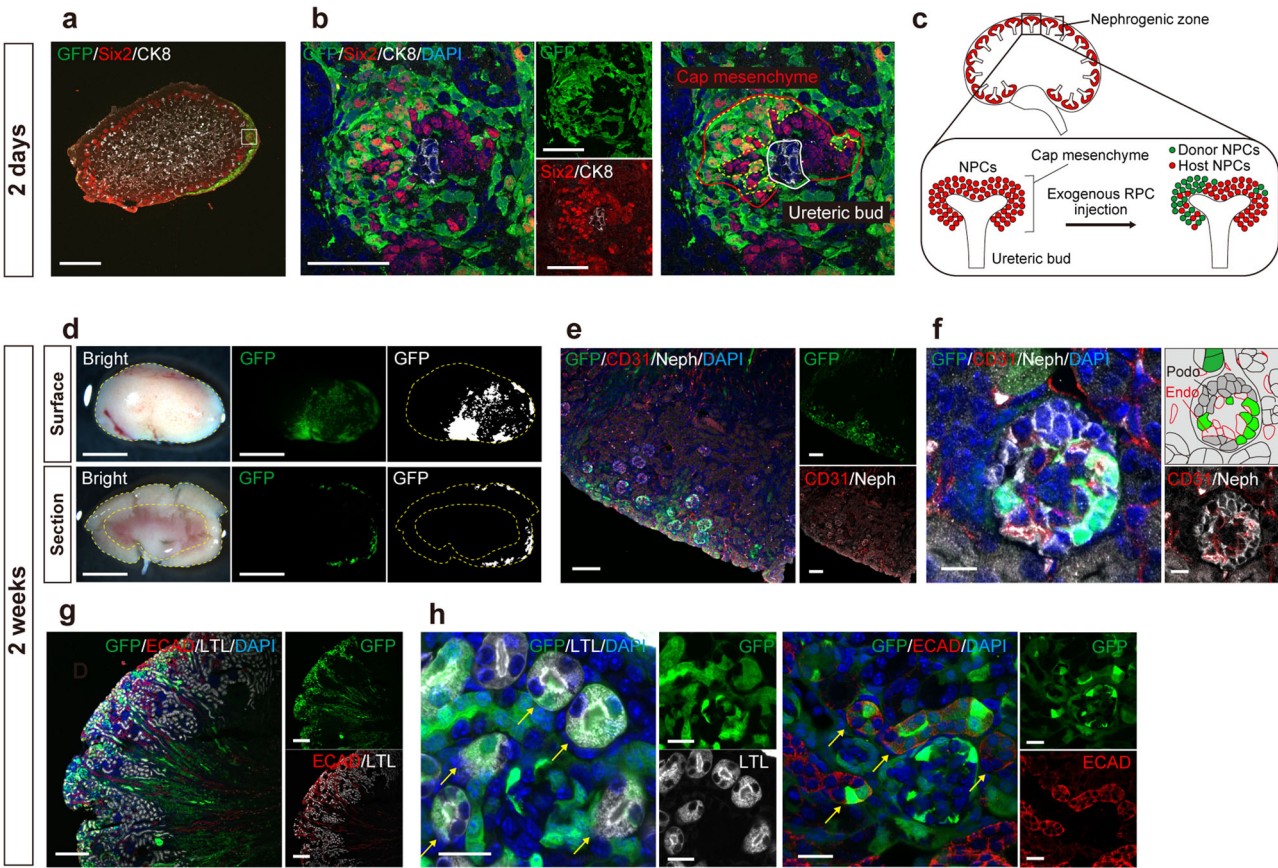

**Fig. 2 Successful integration and differentiation of donor cells in the host neonatal nephrogenic niche. a, b** Immunostaining of the chimeric niche in the neonatal kidney 2 days after injection of renal progenitor cells (RPCs) (P3.5). In the outermost layer, the cap mesenchyme consisted of a mixture of host nephron progenitor cells (NPCs, Six2 +, GFP −) and donor NPCs (Six2 +, GFP +) surrounding the host ureteric bud (CK8 +, GFP −). **c** A schematic of (**a**) and (**b**). **d** Fluorescence stereomicroscopic images of the surface and the longitudinal section of host kidney 2 weeks after injection. The yellow dotted lines encircle the host renal cortex. **e–h** Immunostaining of (**d**). **e, f** Chimeric glomeruli containing exogenous podocytes (Neph +, GFP +), nourished by the host endothelial cells (CD31 +, GFP −). **g, h** Chimeric proximal tubules (LTL +) and distal tubules (ECAD +) indicated by yellow arrows. Scale bars, 500 μm in (**a**), 50 μm in (**b**), 2 mm in (**d**), 100 μm in (**e**), 10 μm in (**f**), 200 μm in (**g**), and 20 μm in (**h**). CK8, cytokeratin 8; DAPI, 4′,6-diamidino-2-phenylindole; ECAD, E-cadherin; Endo, endothelial cell; GFP, green fluorescent protein; LTL, lotus tetragonolobus lectin; Neph, Nephrin; Podo, podocyte; Six2, sine oculis homeobox homolog 2.

**Repeated drug administration study using the long-term viable chimeric nephrons.** Taking advantage of the long-term viability of chimeric nephrons, a chronic toxicity model was established by administering repeated low doses of CIS (Fig. 6a). With CIS treatment, Kim1 expression in exogenous PTCs dose-dependently increased, comparable with that of the host (Fig. 6b–d). scRNA-seq

comparing 2 months, CIS (−) and 2 months, CIS (+) further confirmed that the extent of CIS-induced changes in gene expression was significantly correlated between host and donor cells (Figs. 3a and 6e). Several genes were commonly upregulated or downregulated in both host and donor cells (Fig. 6f). Among these, *Gclc*[31] and *Gas6*[32] have been reported to play a role in kidney injury recovery (Fig. 6g).

**Table 2 The chimerism rate in the host's left kidney two weeks after the injection of donor mouse renal progenitor cells.**

| Percentage of GFP-positive areas on the surface (%) | Percentage of GFP-positive areas in the cortical region on the longitudinal section (%)[a] |
|---|---|
| 17.6 ± 1.2 | 10.2 ± 1.7 |

Mean ± SEM (n = 4 biologically independent samples). The percentage of GFP-positive areas was calculated from fluorescence stereomicroscopic images. Representative images are shown in Fig. 2d.
[a]Calculation was performed after excluding the areas with reddish medullary rays.

**EGFP antigen was accepted by neonatal mice**. Although both the host and donor were B6-based inbred mice, the donor cells expressed EGFP, a minor histocompatibility antigen that activates cytotoxic T cells[33]. To assess whether T cell-mediated rejection occurs, the neonatal chimeric nephrons were compared with adult transplantation models, in which EGFP-positive mouse RPC were transplanted under the renal capsule of adult female B6 or NOG mice (Supplementary Fig. 4a). After 2 weeks, a large infiltration of CD3-positive T lymphocytes was observed in adult B6 mice, whereas there was minimal infiltration in adult NOG and neonatal B6 mice (Supplementary Fig. 4b–d). In addition, in adult B6 mice without any nephrotoxic drugs, PTCs in the spheroids showed a high expression of Kim1 (Supplementary Fig. 4e, f). These observations suggest that neonatal mice are less likely to reject the EGFP antigen compared to adult mice.

**Rat-to-mouse xenogeneic chimeric nephrons in neonatal immunocompromised mice**. RPCs from Sprague-Dawley-Tg (CAG-EGFP) rats (EGFP rats) were injected under the renal capsule of neonatal B6 mice. After 5 days, chimeric tubular structures were observed (Supplementary Fig. 5a, b). After 2 weeks, however, no notable structures remained and a substantial infiltration of CD3-positive cells was observed, indicating T cell-mediated rejection (Supplementary Fig. 5c, d). Thereafter, rat RPCs were injected to neonatal NOG mice (Fig. 7a). After 2 weeks, chimeric nephrons formed in all hosts (n = 13) and reproduced CIS-induced toxicity in exogenous rat PTCs (Fig. 7b–d). Furthermore, the chimeric nephrons remained viable even after 2 months, indicating their long-term survival (Fig. 7e, f).

**Generation of human nephrons in the neonatal mouse kidneys**. Human NPCs were induced from EGFP-labeled hiPSCs following previous reports (Fig. 8a)[34,35]. Among the induced cells, 72% were Itga8-positive NPCs (n = 12, Fig. 8b). The in vitro differentiation ability of the NPCs was confirmed prior to injection (Fig. 8c, d). Initially, the human NPCs were injected into neonatal NOG mice after enzymatic dissociation of NPC spheres; however, the cells disappeared within 7 days, leaving nonspecific connective tissue. Two main reasons were hypothesized for this: the single-cell state triggered cell-death[36–38], and Wnt signals secreted by ureteric buds in fetal kidneys that promote NPC epithelialization[39] were inadequate in host neonatal kidneys[40]. To address the first problem, NPC spheres were broken up by pipetting into clusters (Supplementary Fig. 6a). As for the second problem, we exploited the fact that fetal mouse spinal cords that secrete Wnt signals can promote NPC epithelialization by contact for >24 h[34,41,42]. Since bulk spinal cords cannot be transplanted under the neonatal renal capsule, enzymatically dissociated spinal cord cells were co-injected with human NPC clusters. In vitro validation confirmed that spinal cord cells dissociated and re-aggregated can still promote epithelialization of human NPCs (Supplementary Fig. 6b–d) and those re-aggregated together with human NPCs did not inhibit human nephron formation (Supplementary Fig. 6e–g). With these modifications, the injected human NPCs differentiated into nephrons including glomeruli and tubules in 2 weeks (Fig. 8e–h). The rate of successful nephron formation was significantly lower than allogenic chimera (32 vs. 100%, p < 0.0001, Table 1 and Supplementary Table 1). Human glomeruli were supplied with host mouse vessels (Fig. 8e). Human tubules expressed CTR1 and LTL, although to a lesser extent than the host mouse (Fig. 8f, g). Nonetheless, LTL-positive human PTCs exhibited the expression of human-specific Kim1 in response to CIS administration (Fig. 8g, h).

## Discussion

In this study, we successfully generated long-term viable allo- and xenogeneic chimeric nephrons utilizing the renal developmental niche of neonatal mice. These chimeric nephrons exhibited comparable maturation levels to the host nephrons and demonstrated longevity and functionality. In contrast, mouse renal organoids cultured in vitro remained immature compared to native kidneys, and those transplanted into adult kidneys did not survive long-term. In addition, we found that human NPCs could differentiate into nephrons and replicate drug-induced PTC damage when injected as NPC clusters along with mouse spinal cord cells in neonatal mouse kidneys.

The NNI method used in this study proved to be more practical and accessible compared to other methods for generating chimeric nephrons. Blastocyst complementation requires fine microinjection technique[9], while the *exo utero* method involves delicate manipulation using a three-axis manipulator[11]. Both methods also require a second operation unlike the NNI method: embryo transfer to pseudopregnant mice in blastocyst complementation and cesarean section in the *exo utero* method (Table 1). The survival rate of NNI was satisfactory, especially when using P1.5 neonates, although injured neonates were susceptible to maternal killing[43]. To minimize harm, we made the wounds inconspicuous with fine threads and warmed the neonates in a container scented with the mother before returning them to the cages. Importantly, the formation of chimeric kidneys reached 100% due to improved visualization of the injection site in the host kidney during the procedure (Table 1).

It is expected that chimeric nephrons generated with gene-edited rodent NPCs could provide more straightforward disease models compared to creating genetically modified mouse lines. This approach holds promise for studying congenital renal diseases such as Alport syndrome, polycystic kidney disease, or nephronophthisis, as well as drug-induced PTC toxicity, by modifying single causative genes or multiple PTC transporters. NPCs can be cultured undifferentiated for an extended period to facilitate gene editing and maintenance[19]. In addition, host neonatal mice may accept NPCs expressing exogenous proteins. In this study, neonatal mice exhibited tolerance to the EGFP antigen, which triggers a T-cell immune response, supporting the notion that their immune system is less mature compared to adults[33,44].

This study has several limitations. First, although human nephrons were generated in neonatal mouse kidneys and replicated CIS-induced PTC damage, their integration with host mouse nephrons, crucial for long-term evaluation, has yet to be established. Moreover, the generation of human nephrons was inconsistent, and their expression of mature markers was weaker than in host mouse nephrons. It is possible that an interspecies barrier exists between human and rodent cells, where host NPCs impede the integration of human NPCs with the host ureteric

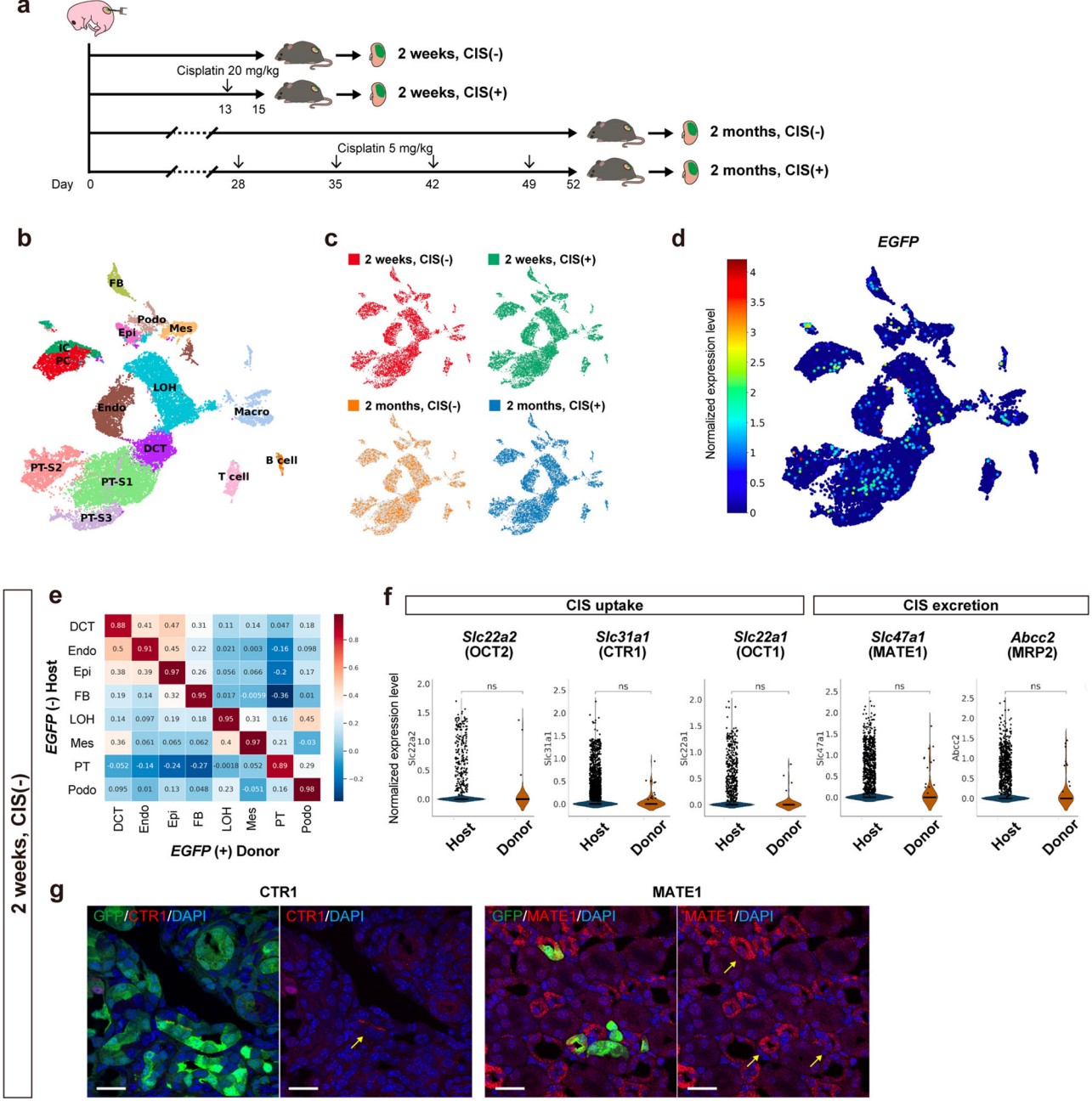

**Fig. 3 Maturation of exogenous nephrons (*EGFP*+) comparable with host nephrons (*EGFP* − ) evaluated with single-cell RNA-sequencing (scRNA-seq). a** A schematic of the sample collection. **b** Uniform manifold approximation and projection (UMAP) displaying unsupervised clustering of all 24,682 cells into 15 distinct, with annotation based on the expressions of previously reported marker genes. **c** Distribution of the four samples in the UMAP. **d** Distribution of *EGFP* (+) donor cells in the UMAP. The color scale indicates log-normalized *EGFP* expressions in each cell. **e** Pearson correlation analysis of the 2 weeks, CIS (−) sample, illustrating the correlation between host (*EGFP* −, n = 6141 cells) and donor (*EGFP* +, n = 64 cells) cells regarding the expression patterns of each cell type. **f** Violin plots depicting normalized expression levels of transporters involved in CIS uptake (*Slc22a2*, *Slc31a1*, and *Slc22a1*) and excretion (*Slc47a1* and *Abcc2*) in host (n = 1291 cells) and donor (n = 20 cells) proximal tubule cells of the 2 weeks, CIS (−) sample. Data were analyzed using Welch's t-test. **g** Immunostaining displaying the expressions of CTR1 and MATE1 in chimeric nephrons, indicated by yellow arrows. Scale bars, 20 μm in (**g**). CIS, cisplatin; DAPI, 4',6-diamidino-2-phenylindole; DCT, distal convoluted tubule; EGFP, enhanced green fluorescent protein; Endo, endothelial cell; Epi, epithelial cell; FB, fibroblast; GFP, green fluorescent protein; IC, collecting duct intercalated cell; LOH, loop of Henle; Macro, macrophage; Mes, mesangial cell; NK, natural killer cell, ns, not significant; PC, collecting duct principal cell; Podo, podocyte; PT, proximal tubule; PT-S1 ~ 3, PT-segments 1–3; RBC, red blood cell.

buds. In addition, induced progenitor cells may have low viability and chimera-forming ability, as indicated by the necessity for modifications before injection in this study. To generate human chimeric nephrons, it might be beneficial to utilize an animal model in which host NPCs are selectively eliminated[12] or to

explore strategies such as identifying factors that enhance the chimera-forming ability of human NPCs. Nonetheless, even without a connection to the host, these in vivo human nephrons may still serve a valuable model for drug screening and pathological analysis, particularly within a limited timeframe. Existing

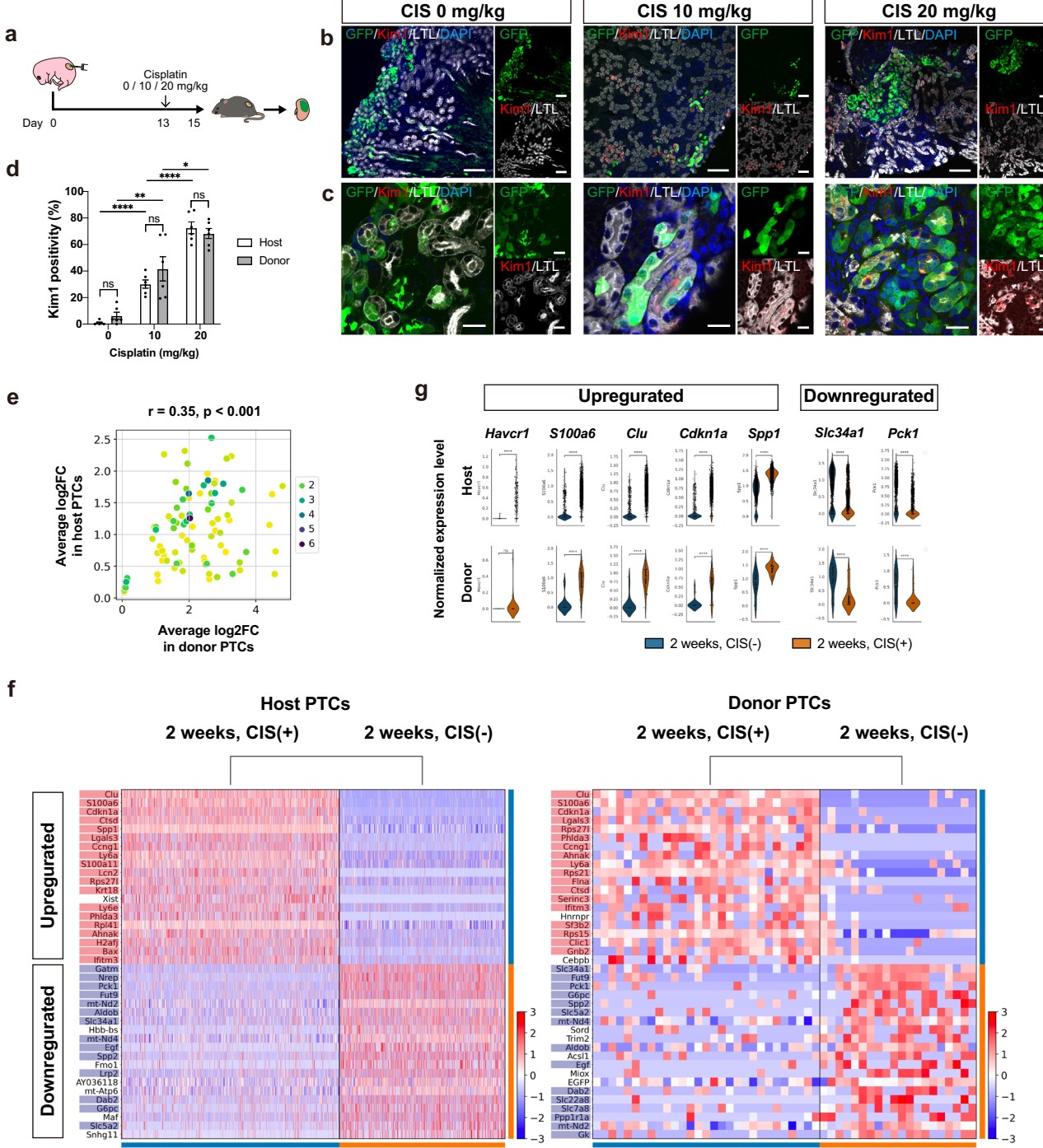

**Fig. 4 CIS-induced acute kidney injury model using chimeric nephrons. a** A schematic of the CIS-induced acute kidney injury model. **b**, **c** Immunostaining images of chimeric nephrons. The expression of kidney injury molecule 1 (Kim1) in proximal tubule cells (PTCs, LTL+) of host (GFP−) and donor (GFP+) origin increases in a CIS dose-dependent manner. **d** Kim1 positivity of host and donor PTCs (n = 6 sections from 3 biologically independent samples) after CIS treatment. Error bars represent means ± SEM. Data were analyzed using the two-tailed unpaired t-test. ns, not significant. *p < 0.05; **p < 0.01; ****p < 0.0001. **e**–**g** Comparisons between the samples at 2 weeks, CIS (−) and 2 weeks, CIS (+) (Fig. 3a) for both host and donor PTCs. **e** A comparison of the extent of individual gene expression changes upon CIS administration, shown by log2 fold change (log2FC), between host PTCs (n = 1291 cells in 2 weeks, CIS (−) vs. n = 1698 cells in 2 weeks, CIS (+)) and donor PTCs (n = 20 cells in 2 weeks, CIS (−) vs. n = 29 cells in 2 weeks, CIS (+)). **f** Heatmaps displaying genes with high variability, both up- and downregulated upon CIS administration, in host and donor PTCs. Genes with significant expression changes (log2FC > 1 and p < 0.05) in both host and donor PTCs are highlighted in red (upregulated) and blue (downregulated). **g** Violin plots showing normalized expression levels of representative variable genes in host and donor PTCs without (blue) and with (orange) CIS treatment. Scale bars, 100 μm in (**b**) and 20 μm in (**c**). CIS, cisplatin; DAPI, 4′,6-diamidino-2-phenylindole; GFP, green fluorescent protein; LTL, lotus tetragonolobus lectin.

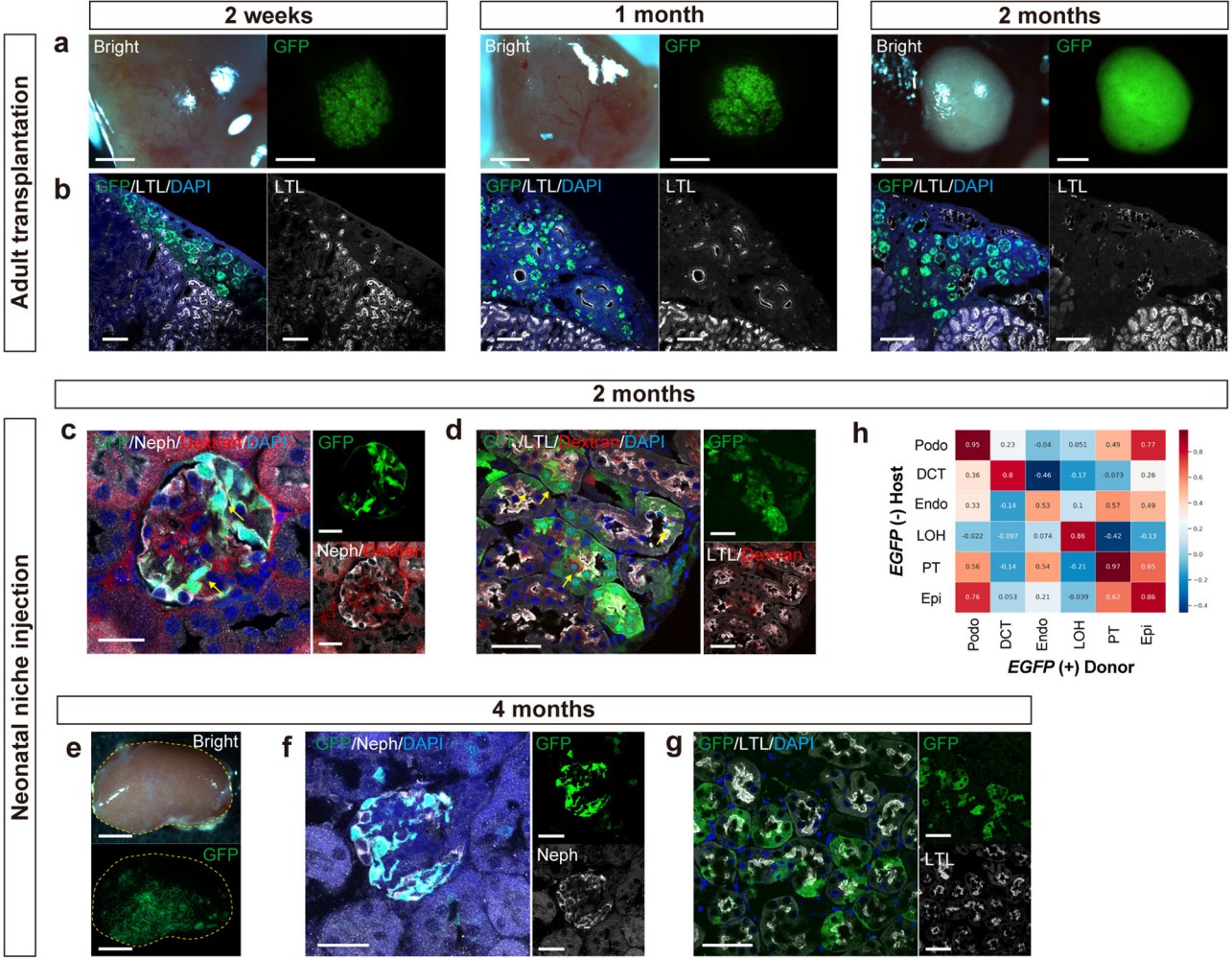

**Fig. 5 Long-term viability and excretion function of chimeric nephrons. a, b** Fluorescence stereomicroscopic images (a) and immunostaining (b) of renal spheroids from EGFP mouse renal progenitor cells transplanted under the kidney capsule of adult NOD/Shi-scid, IL-2RgKO Jic mice (NOG mice) at 2 weeks, 1 month, and 2 months after transplantation. **c, d** Immunostaining of chimeric nephrons 2 months after injection, after systemic administration of fluorescent-labeled low-molecular-weight dextran. Yellow arrows indicate dextran excretion (**d**) and reabsorption (**e**) by the chimeric nephrons.
**e** Fluorescence stereomicroscopic images of the host kidney 4 months after injection. **f, g** Immunostaining of chimeric nephrons 4 months after injection.
**h** Pearson correlation analysis of the 2 months, CIS (−) sample showing the correlation between host (EGFP −, $n = 1829$ cells) and donor (EGFP +, $n = 10$ cells) regarding the expression patterns of each cell type. Scale bars, 1 mm in (**a**), 100 μm in (**b**), 20 μm in (**c**), 50 μm in (**d**), (**f**), (**g**), and 2 mm in (**e**). DAPI, 4′,6-diamidino-2-phenylindole; EGFP, enhanced green fluorescent protein; GFP, green fluorescent protein; LTL, lotus tetragonolobus lectin; Neph, Nephrin.

animal models have limited predictive value and current in vitro human kidney models are not fully matured[3,4,45]. Recently, a study validated podocytopathy using human renal organoids transplanted into adult mouse kidneys[46], but tubular toxicity have not been evaluated in in vivo human nephrons. Furthermore, efforts to generate human chimeric nephrons within the animal nephrogenic niche holds the potential to eventually lead to the development of transplantable kidneys[47].

As a second limitation, due to the low chimerism rate within the host kidneys, we did not conduct functional assessments employing blood and urine tests. The injected RPCs primarily remained localized to the injection site, resulting in GFP-positive areas comprising less than 20% of the cortex. However, this study demonstrated the feasibility of assessing a small number of donor nephrons through scRNA-seq.

Third, in scRNA-seq where donor cells were defined by *EGFP* gene expression, *EGFP* might not be detected in some donor cells, leading to misclassification as host cells. However, we assume this effect is limited since host cells are more abundant than donor cells, as shown in Fig. 2d and Supplementary Fig. 2a, b. Advancements in

analytical methods such as spatially resolved transcriptomics[48] are expected to enable more accurate cell differentiation.

Lastly, the assessment of drug-induced injury through scRNA-seq and immunostaining has primarily concentrated on proximal tubules. Further research is necessary to assess the pathophysiology of glomeruli and other tubular segments.

In conclusion, chimeric nephrons derived from progenitor cells were generated in neonatal mice and showed comparable maturation levels as the host nephrons. They were viable for a long period and applicable to acute and chronic drug testing. Human nephrons were also generated in the neonatal kidneys and reproduced drug-induced toxicity. This NNI method holds great promise as an in vivo approach for chimeric nephron generation, with potential applications in kidney regeneration, drug screening, and pathological analysis.

## Methods

**Research animals.** All animal experiments were performed following the Guidelines for the Proper Conduct of Animal

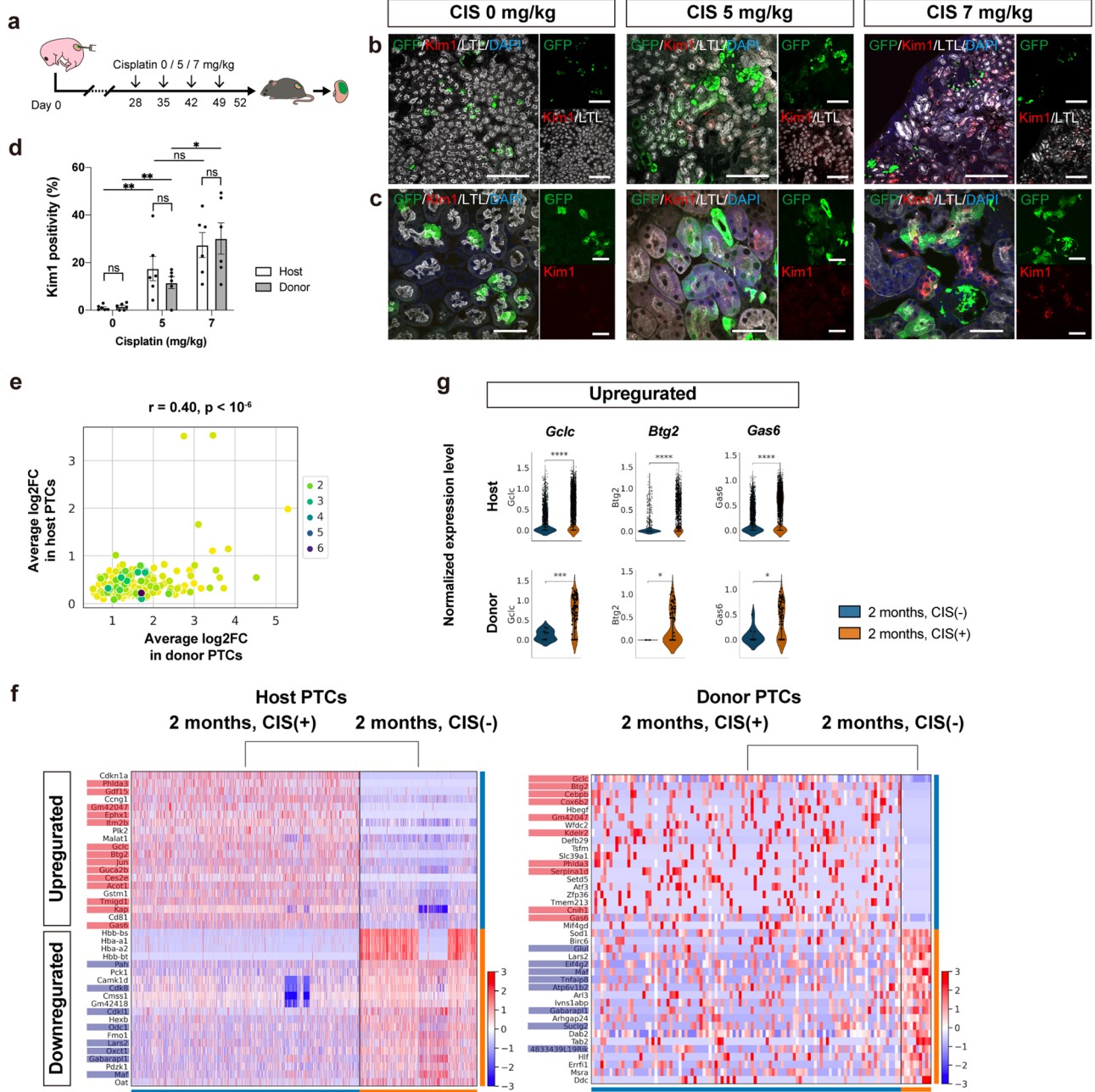

**Fig. 6 Application of chimeric nephrons in repeated-dose testing. a** A schematic of the repeated-dose testing of CIS. **b**, **c** Immunostaining demonstrating that the expression level of Kim1 in proximal tubule cells (PTCs, LTL+) of the host (GFP−) and donor (GFP+) origin increases in a dose-dependent manner with CIS treatment. **d** Kim1 positivity of host and donor PTCs after CIS treatment ($n = 6$ sections from 3 biologically independent samples). Error bars represent means ± SEM. Data were analyzed using the two-tailed unpaired t-test. ns, not significant. *$p < 0.05$; **$p < 0.01$. **e**–**g** Comparison between the samples at 2 months, CIS (−) and 2 months, CIS (+) (Fig. 3a) for both host and donor PTCs. **e** A comparison of the **e** extent of individual gene expression changes upon repeated CIS administration, shown by log2 fold change (log2FC), between host PTCs ($n = 1829$ cells in 2 months, CIS (−) vs. $n = 3559$ cells in 2 months, CIS (+)) and donor PTCs ($n = 10$ cells in 2 months, CIS (−) vs. $n = 103$ cells in 2 months, CIS (+)). **f** Heatmaps displaying genes with high variability, both up- and downregulated upon CIS administration, in host and donor PTCs. Genes with significant expression changes (log2FC >1 and $p < 0.05$) in both host and donor PTCs are highlighted in red (upregulated) and blue (downregulated). **g** Violin plots showing normalized expression levels of representative variable genes in host and donor PTCs without (blue) and with (orange) CIS treatment. Scale bars, 200 μm (**b**) and 50 μm in (**c**). CIS, cisplatin; DAPI, 4′,6-diamidino-2-phenylindole; GFP, green fluorescent protein; LTL, lotus tetragonolobus lectin; Kim1, kidney injury molecule 1.

Experiments of the Science Council of Japan (2006) and were approved by the Institutional Animal Care and Use Committee of the Jikei University School of Medicine (approval numbers: 2021-072 and D2021-070). Pregnant female B6 mice, EGFP mice, EGFP rats, and adult female B6 mice were purchased from SLC. Adult female NOG mice were purchased from CLEA and used at

8–16 weeks of age. All mice were maintained on a 12-h light/dark cycle with free access to standard diet and water.

**Dissociation of EGFP-labeled fetal kidneys to obtain RPCs.** Rodent RPCs were dissociated from EGFP-labeled fetal kidneys of

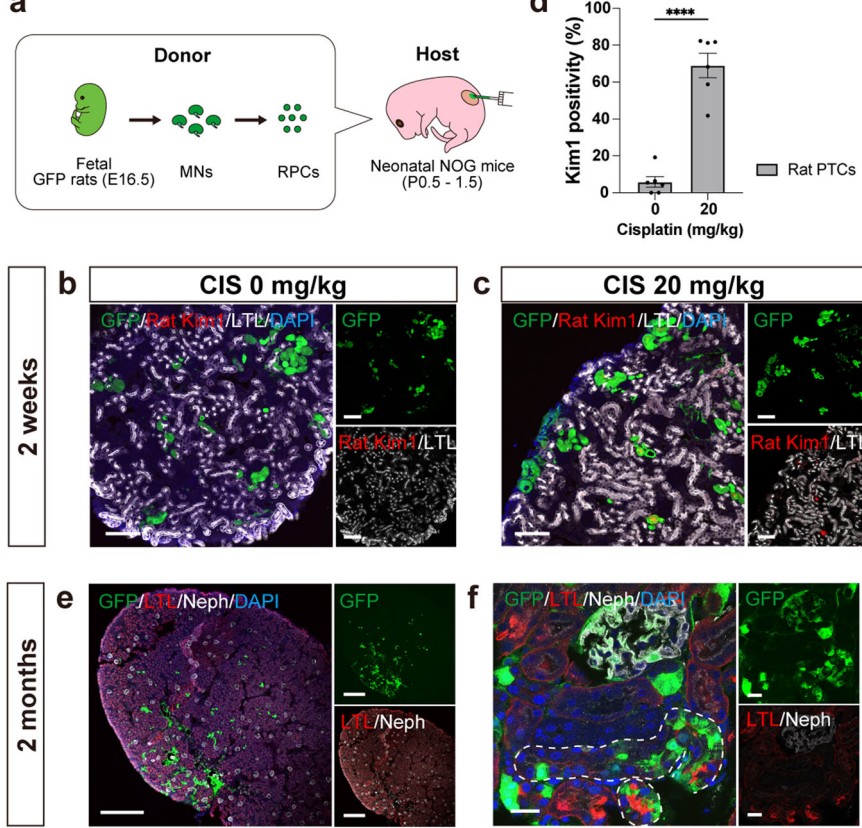

**Fig. 7 Xenogeneic chimeric nephrons derived from rat renal progenitor cells (RPCs) in immunosuppressed neonatal mouse hosts. a** A schematic of the method to inject rat RPCs under the renal capsule of neonatal NOD/Shi-scid, IL-2RgKO Jic mice. **b**, **c** Immunostaining for rat-specific Kim1 in exogenous rat proximal tubules after 2 weeks, comparing samples without CIS treatment (**b**) and with CIS treatment (**c**). **d** Kim1 positivity of exogenous rat proximal tubule cells ($n = 6$ sections from 3 biologically independent samples) after CIS treatment. Error bars represent means ± SEM. Data were analyzed using the two-tailed unpaired t-test. ****$p < 0.0001$. **e**, **f** Immunostaining of xenogeneic chimeric nephrons after 2 months showing chimeric glomeruli (Neph+) and proximal tubules (LTL +, white dotted lines). Scale bars, 100 μm in (**b**) and (**c**), 500 μm in (**e**), and 20 μm in (**f**). CIS, cisplatin; DAPI, 4',6-diamidino-2-phenylindole; GFP, green fluorescent protein; LTL, lotus tetragonolobus lectin; Neph, Nephrin; Kim1, kidney injury molecule 1; PTC, proximal tubule cell.

E14.5 EGFP mice, E16.5 EGFP rats, or E14.5 B6 mice. Fetal kidneys were extracted using micro-tweezers, collected into a 1.5-mL tube containing 1 mL of Accutase (Innovative Cell Technologies), and vortexed for 30 s. The kidneys were incubated at 37 °C for 5 min and vortexed abruptly, and this process was repeated three times for a total of 15-min incubation time. Finally, gentle manual pipetting using a 200-mL pipette tip was performed to create a single-cell suspension before centrifugation at $300 \times g$ for 5 min. Pellets were resuspended in 1 mL of Minimum Essential Medium α (MEM α; Invitrogen) supplemented with 20% fetal bovine serum (FBS; HyClone Laboratories, Inc.) and 1% antibiotic–antimycotic solution (Thermo Fisher Scientific) and dissociated by gentle manual pipetting. Single-cell suspensions of RPCs were passed through a 40-μm cell strainer (BD Falcon) and centrifuged at $700 \times g$ for 3 min. The supernatant was removed completely, and the tubes were tapped to mix the pellet and incubated on ice for up to 2 h before use.

**Dissociation of spinal cords from fetal mice.** Spinal cords were collected from E13.5–14.5 fetal B6 mice, cut into small pieces with micro-tweezers, and then collected into a 1.5-mL tube containing 1 mL of Accutase. The tubes were incubated at 37 °C for 15 min with vorticing at the beginning (0 min) and end (15 min) of incubation and then centrifuged at $1000 \times g$ for 3 min. Pellets were resuspended in 1 mL of MEM α with 20% FBS and 1% antibiotic–antimycotic solution and dissociated by gentle manual

pipetting. Single-cell suspensions of spinal cord cells were passed through a 40-μm cell strainer and centrifuged at $1000 \times g$ for 3 min. For co-injection with human NPCs, the supernatant was removed completely and the tubes were tapped to mix the pellet and incubated on ice for up to 2 h until use. For in vitro culture of spinal cord cell spheres, $2 \times 10^5$ cells/200 μL/well resuspended in 1 mL of MEM α with 20% FBS, 1% antibiotic–antimycotic solution, and 10 μM Y27632 were distributed in wells of U-bottom 96-well low-cell binding plates (Sumitomo Bakelite). The plate was centrifuged at 1000 rpm for 4 min and incubated overnight at 37 °C in a 5% $CO_2$ incubator for re-aggregation.

**Formation of mouse renal spheroids.** The pellets of MNs from E14.5 fetal B6 or EGFP mice were resuspended in 1 mL of MEM α with 20% FBS, 1% antibiotic–antimycotic solution, and 10 μM Y27632. Each well contained $2 \times 10^5$ cells/200 μL of cell suspensions using the U-bottom 96-well low-cell binding plates. Finally, the plate was centrifuged at 1000 rpm for 4 min and incubated overnight at 37 °C in a 5% $CO_2$ incubator for re-aggregation.

**In vitro culture of mouse renal spheroids.** On the next day (day 1), spheroids from B6 mice were transferred onto the air–liquid interface of a polycarbonate filter with an average pore size of 0.4 μm (Transwell, Corning). The medium consisted of MEM α with 20% FBS and 1% antibiotic–antimycotic solution. The medium

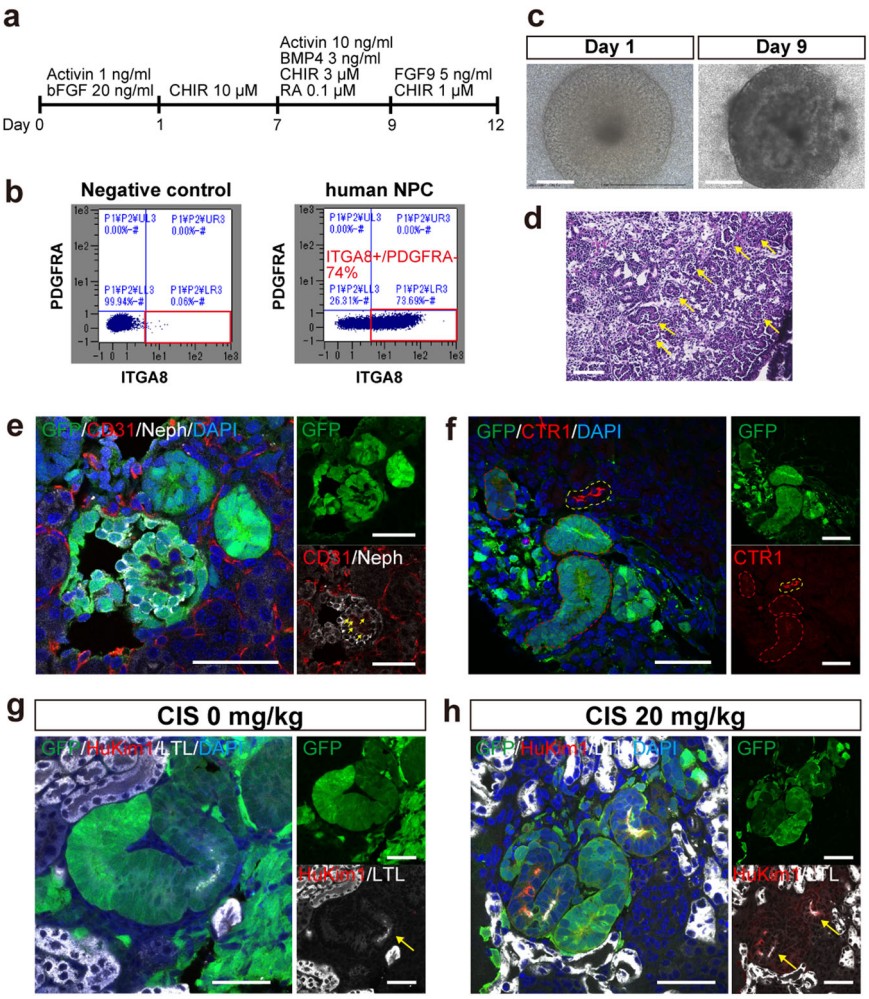

**Fig. 8 Human nephrons generated in immunosuppressed neonatal mouse kidneys. a** A schematic of the protocol followed for inducing nephron progenitor cells (NPCs) from EGFP-labeled human-induced pluripotent stem cells. **b** Representative images of the flow cytometric analysis of NPCs ($n = 12$ sections from 3 biologically independent samples). **c** Microscopic images of the in vitro differentiation of human NPCs for 9 days. **d** Hematoxylin and eosin-stained section of (**c**) demonstrating glomeruli (yellow arrows) and tubules formation. **e–h** Immunostaining of human nephrons in the neonatal kidneys 2 weeks after injection. **e** A human glomerulus (GFP +, Neph+) supplied by host vessels (yellow arrows. GFP−, CD31+). **f** CTR1 expression of human tubules (red dotted lines) and host mouse tubules (a yellow dotted line). **g, h** HuKim1 expression in human proximal tubules (LTL+, yellow arrows), comparing samples without CIS treatment (**g**) and with CIS treatment (**h**). Scale bars, 500 μm in (**c**), 100 μm in (**d**), 50 μm in (**e**), (**f**), (**g**), and (**h**). CIS, cisplatin; DAPI, 4′,6-diamidino-2-phenylindole; EGFP, enhanced green fluorescent protein; HuKim1, human-specific kidney injury molecule 1; LTL, lotus tetragonolobus lectin; Neph, Nephrin.

was changed every other day, and spheroids were collected on days 7 and 12 and subjected to immunostaining and RT-qPCR.

**Transplantation of mouse renal spheroids under the renal capsule of adult mice.** Spheroids from EGFP mice on day 1 were used. The recipient female B6 or NOD/Shi-scid, IL-2RgKO Jic (NOG) mice were anesthetized with isoflurane inhalation, and a midline abdominal incision was made. The intestine was moved to the side to expose the kidney, and the capsule in the lower part of the kidney was dissected at nearly 1 mm using a microshear. The tip of the outer cylinder of the 22 G Surflo I.V. Catheter (Terumo) was cut at an angle. The outer cylinder was inserted through the incision in the renal capsule with its cut surface facing the renal parenchyma, and to detach the renal capsule from the parenchyma, a small amount of saline was placed under the renal capsule. A spheroid was aspirated into the outer cylinder of the Surflo and was inserted under the renal capsule through the incision. The abdominal incision was then closed with a 5-0 thread. On days 14, 28, and 56 after transplantation, recipient

mice were euthanized, and the transplanted spheroids were collected.

**Maintenance of hiPSCs.** EGFP-labeled hiPSC line (317-12-Ff)[49] was maintained on iMatrix-511 (Nippi)-coated 6-well plate (IWAKI) in StemFit AK02N medium (Ajinomoto). Cells were cultured at 37 °C in a humidified atmosphere of 5% $CO_2$ and passaged every 7 days. On each passage day, 0.5× TryPLE Select (Thermo Fisher Scientific) was used for detachment and then seeded at $0.65 \times 10^4$ cells/well in the 6-well plate. The cell culture medium was replaced on days 2, 4, and 5.

**Induction of human NPCs from hiPSCs.** NPCs were induced from hiPSCs according to established methods[34,35]. On day 0, iPSCs were re-aggregated in a V-bottom, 96-well, low-cell-binding plate (Sumitomo Bakelite) that contained a basic medium supplemented with 1 ng/mL human activin A (R&D Systems), 20 ng/mL human basic fibroblast growth factor (FGF; R&D Systems), and 10 μM Y27632 (Wako) with 10,000 cells per

aggregation. On day 1, the spheres were transferred to a U-bottom, 96-well, low-cell-binding plate that contained a medium supplemented with 10 μM CHIR (Axon Medchem) and 10 μM Y27632. Half medium change was performed on days 3 and 5 with the medium of the same composition as day 1. On day 7, the spheres were transferred to a U-bottom, 96-well, low-cell-binding plate that contained a medium supplemented with 3 ng/mL human bone morphogenetic protein 4 (R&D Systems), 0.1 μM retinoic acid (Sigma-Aldrich), 10 ng/mL human activin A, 3 μM CHIR, and 10 μM Y27632. On day 9, the spheres were transferred again to a U-bottom, 96-well, low-cell-binding plate that contained a medium supplemented with 5 ng/mL human glia-activating factor 9, 1 μM CHIR, and 10 μM Y27632. On day 12, the spheres were enzymatically dissociated or broken into clusters by manual pipetting 20 times using a P200 tip. For enzymatic dissociation, spheres were collected in 0.5× TrypLE™ Select (Thermo Fisher Scientific) and dissociated into cell suspensions. To eliminate any remaining clumps, the cells were passed through a 40-mm cell strainer and centrifuged at $300 \times g$ for 3 min. After the supernatant was removed, the tubes were tapped to mix the pellet, incubated on ice, and used within 2 h.

**Flow cytometry of human NPCs**. To evaluate the percentage of NPCs in day 12 NPC spheres, flow cytometry was performed. Biotinylated anti-ITGA8 (R&D Systems), allophycocyanin-conjugated streptavidin (BioLegend), and phycoerythrin-conjugated anti-PDGFRA (BioLegend) were used for cell staining. The allophycocyanin (Itga8)-positive and phycoerythrin (PDGFRA)-negative fraction was designated as the NPC population. The cut-offs were determined using a negative control without the primary antibodies. A minimum of $5 \times 10^5$ cells was secured in each experiment.

**In vitro differentiation of human NPCs by chemical induction**. To check the differentiation capacity of NPCs from each induction, day 12 NPC spheres were transferred onto Transwell and cultured in KR5 medium (DMEM/F12 [(1:1) (1×), Thermo Fisher Scientific] with 5% KnockOut™ Serum Replacement [Thermo Fisher Scientific]) supplemented with 3 μM CHIR and 200 ng/mL human basic FGF for 2 days. The medium was changed to KR5 medium on day 2 and then changed every other day. The spheres were collected on day 9 and evaluated in hematoxylin–eosin (HE) staining.

**In vitro differentiation of human NPCs co-cultured with spinal cords or spinal cord cell spheres**. The cut spinal cord from fetal B6 mice (E13.5–14.5) or the spinal cord cell sphere that was re-aggregated overnight was placed next to the day 12 NPC sphere on Transwell. They were cultured in MEM α with 20% FBS and 1% antibiotic–antimycotic solution, and the medium was changed every other day until collection.

**In vitro differentiation of human NPCs mixed with spinal cord cells**. $1 \times 10^5$ cells of the day 12 human NPCs and $1 \times 10^5$ cells of fetal mouse spinal cord cells/200 μL/well were resuspended in MEM α with 20% FBS, 1% antibiotic–antimycotic solution, and 10 μM Y27632 and distributed to wells of U-bottom 96-well low-cell binding plates. The plate was centrifuged at 1000 rpm for 4 min and incubated at 37 °C overnight. On the next day, the spheres were placed on Transwell and cultured in MEM α with 20% FBS and 1% antibiotic–antimycotic solution, and the medium was changed every other day until collection.

**NNI of rodent RPCs and human NPCs with/without spinal cord cells**. P0.5-1.5 (E19.5–20.5 equivalent) neonates from B6 or NOG mice were taken out from the cages with their mothers and anesthetized with isoflurane. Subsequent procedures were conducted under a fluorescence stereomicroscope (M205FA; Leica Microsystems). A vertical incision was made on the left side of the spine. To expose the left kidney, light pressure was applied with micro-tweezers (11253-25; Dumont) from both sides of the incision. A suspension of dissociated RPCs was aspirated into a 34 G Hamilton syringe (Saito Medical Instruments Inc.), and approximately 1 μL was injected under the renal capsule (approximately $1 \times 10^6$ cells). Injected cells were confirmed by EGFP fluorescence. Any leaked suspension was wiped up with Hanks' balanced salt solution using a cotton swab, then the kidney was returned to the body and the incision was sutured with 8-0 nylon (Muranaka Medical Instruments). Following the surgery, neonates were identified by finger or tail cuts and placed on heated plates at 37 °C until regaining consciousness. Once awake, they were returned to their original cages and reared by their biological mothers.

**Systemic administration of cisplatin**. CIS (Fujifilm Wako) was diluted in saline to make a 0.5 mg/mL solution. Then the host mice were then injected intraperitoneally with 5–20 mg/kg of the solution using a syringe with a 29 G needle (Terumo).

**Systemic administration of fluorescent-labeled low-molecular-weight dextran**. Eight weeks after the injection, the host mice were injected with 15 mg/kg of tetramethylrhodamine-conjugated, 10-kDa, low-molecular-weight dextran (D1817, Thermo Fisher Scientific) into the inferior vena cava twice under general anesthesia. The first and second injections were timed 5 min apart. Ten minutes after the first injection, the left kidneys were collected, and the hosts were immediately euthanized by phlebotomy.

**Immunostaining and HE staining of frozen sections**. Specimens were fixed in 4% paraformaldehyde (Wako) in phosphate-buffered saline (PBS) overnight and dehydrated in 15% sucrose in PBS overnight and in 30% sucrose in PBS overnight at 4 °C. Specimens were embedded in an OCT compound (Sakura Finetek), and 10-μm thick-frozen sections were prepared. HE staining was performed according to the standard procedures. Antigen retrieval for immunostaining was performed in 10% HistoVT One/PBS (Nacalai Tesque) in a warm bath at 70 °C for 20 min. After blocking with Blocking One Histo (Nacalai Tesque) for 10 min at room temperature, the sections were incubated with primary antibodies (Supplementary Table 2) and then with secondary antibodies conjugated with Alexa Fluor 488, 546, or 647 for 1 h at room temperature. Sections were mounted with Pro-Long Gold Antifade Mountant with 4',6-diamidino-2-phenylindole (DAPI) (Thermo Fisher Scientific). Each sample was examined under a fluorescence microscope (LSM880 confocal).

**Whole-mount immunostaining**. Cultured renal spheroids were fixed with 4% paraformaldehyde in PBS for 15 min at 4 °C, washed three times with PBS. Samples were blocked using 1% donkey serum, 0.2% skimmed milk, and 0.3% Triton X/PBS for 1 h at room temperature, and incubated overnight at 4 °C with primary antibodies. After washing three times with PBS, the samples were incubated with secondary antibodies for 1 h at room temperature. Samples were mounted with ProLong Gold Antifade Mountant with DAPI. Each sample was examined under LSM880 confocal.

**Measurement of the Kim1 positivity rate of proximal tubule cells (PTCs)**. Immunostaining images of chimeric nephrons with

or without cisplatin administration collected 2 weeks and 2 months after cell injection and renal spheroids transplanted to adult B6 and NOG mice collected after 2 weeks were acquired with a confocal microscope. The number of total host (GFP −) and donor (GFP +) PTCs and kidney injury molecule 1 (Kim1)-positive host and donor PTCs were counted manually. The Kim1 positivity rates of the host and donor PTCs were calculated in two images of the same magnification of randomly selected sections (using ×20 lens, 0.72-mm square) from three host mice. A total of six images were analyzed in each group. In this study, two investigators performed cell counting.

**Measurement of the CD3 positivity rate**. Immunostaining images of chimeric nephrons regenerated from RPCs injected into neonates and renal spheroids transplanted to adult B6 or NOG mice collected after 2 weeks were acquired with a confocal microscope. The number of CD3-positive cells was measured manually in two images of the same magnification of randomly selected sections (using ×63 lens, 0.22-mm square) from three host mice. A total of six images were analyzed in each group. Two investigators performed cell counting in this study.

**Single-cell preparation for scRNA-seq**. Four types of samples were prepared: 2 weeks after injection without CIS administration (2 weeks, CIS (−)) from 4 host mice, 2 weeks after injection with a single 20 mg/kg CIS administration 2 days before collection (2 weeks, CIS (+)) from 4 host mice, 2 months after injection without CIS administration (2 months, CIS (−)) from 7 host mice, and 2 months after injection with weekly 5 mg/kg CIS administration for four weeks with the last dose administered 3 days before collection (2 months, CIS (+)) from 6 host mice.

Soon after the host mice were euthanized by cervical dislocation, thoracotomy was performed, and blood was released by the thoracic aorta dissection. The left kidney was collected, and the renal capsule was stripped in Hanks' balanced salt solution without calcium and magnesium. GFP (+) areas were trimmed under fluorescence stereomicroscope (M205FA; Leica Microsystems), broken into small pieces with a new razor blade, and collected into a 15-mL tube containing 1 mL of PBS. The tubes were centrifuged at $700 \times g$ for 3 min, and the supernatant was removed. Then, 1 mL of Accutase was added into the tubes and vortexed for 30 s, and the tubes were incubated at 37 °C for 15 min. Gentle manual pipetting using a 1000-mL pipette tip was performed, and 1 mL of MEM α with 20% FBS and 1% antibiotic–antimycotic solution were added to stop the reaction. Suspensions were passed through a 40-mm cell strainer to a new 15-mL tube. Afterward, debris were removed using a debris-removal solution (Miltenyi Biotec), according to the manufacture's protocol. After counting the cells, the tubes were centrifuged at $400 \times g$ for 10 min, and the supernatant was removed completely. The pellet was suspended in STEM-CELLBANKER (ZENOGEN PHARMA CO., LTD.) and frozen at −80 °C before use.

**10X Genomics scRNA-seq, library preparation, and sequencing**. The frozen samples were thawed using ThawSTAR (BioLife Solutions) and transferred to 10 mL of 10% FBS in RPMI 1640 media (Thermo Fisher Scientific). After centrifugation, the samples were resuspended in 1 mL of 1% bovine serum albumin (Thermo Fisher Scientific) in PBS. The cell suspensions were filtered through a 40-mm cell strainer and resuspended in PBS with 1% FBS at a stock concentration within the recommended range (typically $0.8–10 \times 10^6$ cells/mL) and loaded at a volume to target 10,000 cells on a Chromium Single-Cell Instrument (10x Genomics) to generate single-cell gel bead-in-emulsions (GEMs).

Single-cell RNA-seq libraries were prepared using Chromium Single-Cell 3′ Library & Gel Bead Kit (10x Genomics). GEM reverse transcription (GEM-RT) was performed to produce a barcoded, full-length cDNA from poly-adenylated mRNA. After incubation, GEMs were broken, the pooled post-GEM-RT reaction mixtures were recovered, and cDNA was purified with silane magnetic beads (DynaBeads MyOne Silane Beads; PN37002D; Thermo Fisher Scientific). The whole purified post-GEM-RT product was amplified by PCR. This amplification reaction generated sufficient material to construct a 3″ cDNA library. The sequencing libraries were prepared following the 10x Genomics Single Cell 3' mRNA kit protocol. Libraries were sequenced on a DNBSEQ-G400 (MGI) according to the manufacturer's guidelines: 3′ v3.1 - 28x8x0x91 [read1xindex1xindex2xread2].

**Sequencing data quality control and preprocessing**. Raw reads were processed by Cell Ranger 6.1.2 (10x Genomics) and run against the Rnor_6.0-104 reference genome for the rat sample and the mm10 for the mouse sample. Low-quality cells were filtered by removing cells with count and gene numbers lower than 500 and 200, respectively. Unhealthy cells were determined and removed based on whether they had more than 50% total mitochondrial and ribosomal genes. Highly probable doublet cells were predicted using scrublet doublet finders and then removed.

**Dimensionality reduction and clustering analysis**. The mitochondrial content and total RNA counts were used as regression variables in the normalization procedure by the log normalization method (scanpy), where 3000 genes were selected as variable features. On the basis of the normalized counts, a harmony integration method was performed. All available genes were kept in the integrated dataset, irrespective of their usage as anchors or not. Integrated counts were scaled and centered for principal component (PC) analysis and dimensional reduction. For each scaled dataset, 50 PCs were calculated using the most variable genes. A shared nearest-neighbor graph of 15 cells was constructed. Cell clusters were identified via the Leiden unsupervised graph method. For data visualization, the nonlinear dimensional reduction method UMAP was used based on different corrected linear methods.

**Identification of marker genes and differentially expressed genes**. Marker gene discovery and differential expression analyses were performed with the function rank_genes_group (scanpy) Wilcoxon and over-estimated t-test, respectively.

**Heatmap showing the correlation between host and donor cells among different cell populations**. We used the scanpy tl.dendrogram and pl.correlation_matrix to perform a pairwise whole transcriptome Spearman correlation analysis between host (*EGFP−*) and donor (*EGFP+*) cell groups for each cell type. We used the seaborn python heatmap command to display the corresponding results.

**Violin plots for transporter gene expression between the host and donor PTCs**. The cellxgene violin plotting function of PTCs between host (*EGFP−*) and donor (*EGFP+*) cell groups was used to visualize the gene expression distribution in selected transporter genes.

**Comparison of gene expression fold-change between CIS (+) and CIS (−) cells on PTCs**. The average log2 fold change was calculated from the differential analysis described above using the seaborn scatter plot command line.

**Gene expression on genes with larger fold change between CIS (+) and CIS (−) shown by heatmap for host and donor cells**. The scanpy pl.rank_genes_groups_heatmap, taking the result from the differential expression described above, was used to plot the top highly differentiated genes between the CIS (+) and CIS (−) groups.

**Violin plots for selected genes between CIS (+) and CIS (−) in host and donor PTCs**. The cellxgene violin plotting function of PTCs between the CIS (+) and CIS (−) groups was used to visualize the gene expression distribution difference of top differentiated selected genes.

**RT-qPCR for in vitro renal spheroids**. RNeasy Plus Micro Kit (Qiagen) was used to isolate RNA from fetal kidneys (E14.5) and spheroids cultured for 7 and 12 days, and RNeasy Plus Mini Kit (Qiagen) was used to isolate RNA from mature kidneys (P9) of B6 mice. A total of 50 ng of RNA was reverse-transcribed into cDNA using the PrimeScript$^{TM}$ RT Reagent Kit with gDNA Eraser (Takara). qPCR was performed using TaqMan$^{TM}$ Gene Expression Master Mix (Thermo Fisher Scientific), Taqman Asssay (Supplementary Table 3, Qiagen), and a thermocycler (Rotor-Gene Q, Qiagen). Glyceraldehyde 3-dehydrogenase was used as a housekeeping gene to normalize the expression levels.

**Statistics and reproducibility**. All data are presented as mean ± standard error of the mean (SEM). Data were analyzed using the two-tailed unpaired t-test. A $p$-value of $<0.05$ was considered statistically significant. Experimental data were analyzed using GraphPad Prism version 8.0 (GraphPad Software, Boston, MA, USA) and Microsoft Excel (Microsoft, Redmond, WA, USA). Details are explained in each figure legend.

**Reporting summary**. Further information on research design is available in the Nature Portfolio Reporting Summary linked to this article.

## Data availability

The scRNA-seq data that support the findings of this study have been deposited in the Gene Expression Omnibus under accession number E-GEAD-607. All other data are available from the corresponding author upon reasonable requests. The source data for the graphs are available in Supplementary Data 1.

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

## Acknowledgements

The authors thank H. Hayashi for experimental/technical assistance. We would like to thank Enago (https://www.enago.jp/) for the English language review and KOTAI Bio-technologies, Inc. (Osaka, Japan) for assistance with scRNA-seq analysis. EGFP-labeled hiPSCs were provided by the iPS Cell Research Institute, Kyoto University. This work was supported by the Japan Agency for Medical Research and Development (AMED; grant no. 22bm0704049h0003 and 22bm1223003h0001), JST FOREST Program (grant no. JPMJFR2011), and the Japan Society for the Promotion of Science (JSPS-KAKENHI; grant no. 19K17756, 21K08288, and 22K20898).

## Author contributions

K.Matsui and S.Y. designed the study. K.Matsui, S.Y., and S.C. carried out experiments, analyzed the data, and wrote the manuscript. N.M. established the protocols for NPC induction, K.Morimoto, Y.K., Y.I., Y.S., T.T., T.F., S.T., K.M., E.K., and T.Y. interpreted the data and revised the manuscript. S.Y., Y.S., and T.Y. acquired funding. T.F., S.T., K.M., E.K., and T.Y. supervised the project. All authors have approved the final version of the manuscript.

## Competing interests

The authors declare no competing interests.
