## [Peer Review File · Communications Biology]

Reviewers' comments:

Reviewer #1 (Remarks to the Author):

This study established a neonatal niche injection (NNI) method, which was harnessed to demonstrate that mouse neonatal kidney (P0.5-P1.5) can incorporate allo- and xenogenic nephron progenitor cells (NPCs) to generate chimeric kidney. The neonatal niche allows allogenic NPC maturation beyond the state that could be achieved by in vitro culture or aggregate culture in vivo. Overall, the quality of data is good. Most of the data can support their conclusions. However, there are some concerns to be solved by the authors before publication.

The comparison of maturation between host and donor nephrons only depends on scRNA-seq. It's better to perform staining to detect protein level and location of maturation markers, like solute channels.

The authors need to provide a calculated ratio of chimerisation when allogenic renal progenitor cells (RPCs) were injected into the neonatal niche.

During kidney development, it's reported juxtamedullary nephrons are generated earlier during development while cortical nephrons are generated and mature at a later stage. Therefore, injection of NPC into neonatal kidney may only generate superficial cortical nephrons. Can the authors test some medullary and cortex makers to clarify whether they observed the allogenic NPC integrated into both types of nephrons?

The injection of human pluripotent stem cell derived NPC patch into host mouse does not present strong evidence: 1) It seems the injected hNPC patch formed their own structure instead of integrating to the host kidney. It's not chimeric kidney; 2) there is a lack result showing the maturation level of hNPC-derived nephrons; 3) why the hNPC-derived PTs did not express LTL?

The application of the NNI method is limited. Firstly, the chimeric efficiency can be too low to demonstrate the phenotype in future application. Secondly, based on the current results present by the authors, it is very challenging to make mouse-human chimeric kidney. It may be helpful to discuss the possibility of using NNI method to study/validate genes that are associated with congenital kidney diseases.

Reviewer #2 (Remarks to the Author):

Here, Matsui and colleagues report the ability to cultivate chimeric nephrons derived from mouse, rat and human (iPSC derived) renal progenitors following their injection into the developing kidneys of neonatal mice at the ~3 day stage. The goal is to explore the ability of the progenitor derivatives to become integrated into the host urinary tract. At the 3 day neonate stage, the mouse kidneys have nephron progenitor cells and integration of transplanted cells is possible, but has only been studied for timepoints of several weeks after the transplantation. Now, longer term integration is shown to be possible within the report, where the authors observed these kidneys 2 or 4 months after the transplantation procedure in neonates. Interestingly, the chimeric nephrons were found to recruit host vasculature as well, and the authors observed the filtration and reabsorption of fluorescently labeled dextran, providing several indicators of functionality. Further, and importantly, the authors found a number of features indicating the ability of the chimeric nephrons to become established and mature in the host kidney through employing single cell RNA sequencing, and that they were superior in this regard compared to nephrons in prolonged renal spheroid cultures. Additionally, the authors found that these chimeric nephrons could be used to model acute (cisplatin) and chronic renal damage (repeated low doses of chemical injury), further indicating their physiological integration with the host

kidney and cardiovascular system. Finally, the authors also report the outcomes of testing rat-to-mouse renal progenitors in neonates and human-to-mouse as well, where the latter were grown to 2 weeks post transplantation.

Overall, this manuscript describes crucial advances in approaches to study nephrogenesis in vivo. The work is well-controlled, the paper is nicely written, and the figures are clear and organized. I have no major or minor recommendations for revision and endorse publication of the work.

Reviewer #3 (Remarks to the Author):

In this study, the authors demonstrated the successful generation of chimeric kidneys by injecting exogenous RPCs (Renal Progenitor Cells) into neonatal mouse kidneys. They assessed the long-term viability, functionality, and response to nephrotoxic insults of these chimeric kidneys. The results showed that the chimeric nephrons formed connections to the host urinary tract and remained viable for an extended period, making them useful for evaluating acute and chronic nephrotoxicity.

The authors also attempted to generate chimeric kidneys with human nephrons, but they were not as successful as in the case of mouse chimeric kidneys. Despite these observed limitations, the manuscript represents a notable step forward in the field of xeno-transplantable kidneys. The findings can serve as a foundation to guide further research aimed at generating robust chimeric human kidneys.

1) In Figure 2a, it appears that the GFP+ve donor cells are found only on one side of the kidney, likely indicating their presence primarily at the injection site. Similarly, in Figure 7d, which represents a 2-month-old chimeric kidney, there are very few localized GFP positive cells visible. The limited distribution of GFP+ve cells in both figures raises questions about the uniformity of the chimeric kidneys generated through RPC injection. It is possible that the injected RPCs did not populate the entire kidney uniformly. Please clarify.

2) In Sup Table 1, number of cells expressing GFP is reducing to 0.7% in 2 months (Cis-ve, normal condition) from the initial 1% suggests a decline in the population of GFP+ve cells over time under normal conditions. However, during cisplatin administration, there is an increase in the number of cells expressing GFP in 2 months. This indicates that cisplatin treatment might have influenced the survival or proliferation of cells (including GFP+ve cells), leading to an increase in their numbers compared to the normal condition. In Sup Figure 2c, Cis-ve (normal condition) 2-month group, fibroblast appears to be the major cell type expressing GFP. In Cis+ (cisplatin-treated) 2-month group, cell types like epithelial cells, macrophages, Intercalating cells, and Collecting Duct Principal cells, are the major cell types expressing GFP. Can you correlate these observations and suggest if the same kind of cellular profile is observed during kidney regeneration or any other explanation?

3) Line 169: "Their tubules got dilated after 1 month and disrupted after 2 months (Fig. 5a, b)". This is not evident from the figure.

Minor

4) Figure 5b: DAPI staining is not visible

5) Line 177: "long-term engraftment (Fig. 5g)".The correct figure number is Fig 5h.

6) Figure 2d: What does the images in the bottom row represent? Please provide these details in the figure legend.

7) Line 189: Figure 6f not refereed in the manuscript.

Point-by-point Responses to the Editor's and the Reviewers' Comments

Date: September 24, 2023

Journal: *Communications Biology*

Manuscript ID: COMMSBIO-23-2197

Type of manuscript:

Manuscript title: Long-term viable chimeric nephrons generated from progenitor cells are a reliable model in cisplatin-induced toxicity

Authors: Kenji Matsui, Shuichiro Yamanaka, Sandy Chen, Naoto Matsumoto, Keita Morimoto, Yoshitaka Kinoshita, Yuka Inage, Yatsumu Saito, Tsuyoshi Takamura, Toshinari Fujimoto, Susumu Tajiri, Kei Matsumoto, Eiji Kobayashi, and Takashi Yokoo

Responses to Reviewers' Comments

We greatly appreciate the editor's and the reviewers' detailed comments and suggestions. We have revised the manuscript, the tables, and the figures to address these points. We believe that they have been improved through these revisions. Below are our point-by-point responses to the comments. Responses to the comments are shown in red in the revised manuscript and in this letter.

Reviewer #1

1) The comparison of maturation between host and donor nephrons only depends on scRNA-seq. It's better to perform staining to detect protein level and location of maturation markers, like solute channels.

Response:

We have demonstrated the expression of CTR1 and MATE1 in chimeric tubular cells through immunostaining on allogenic chimeras two weeks after injection, as depicted in Fig. 3g.

2) The authors need to provide a calculated ratio of chimerisation when allogenic renal progenitor cells (RPCs) were injected into the neonatal niche.

Response:

Thank you for pointing out. It is of highly importance to demonstrate the extent to which the donor cells contribute to the host kidney. We have measured the chimerism rate in the cortical regions both on the surface and in the cortical region on the longitudinal sections (Fig. 2d and Table 2).

3) During kidney development, it's reported juxtamedullary nephrons are generated earlier during development while cortical nephrons are generated and mature at a later stage. Therefore, injection of NPC into neonatal kidney may only generate superficial cortical nephrons. Can the authors test some medullary and cortex makers to clarify whether they observed the allogenic NPC integrated into both types

of nephrons?

Response:

As injecting exogenous NPCs into mature nephrons does not result in chimera formation (Fig. 5a, b, and Toyohara. *Stem Cells Transl Med.* 2015; 4: 980-92.), we believe that only those injected into the superficial cap mesenchyme (CM) of neonatal mouse kidneys can form chimeric nephrons. Some cells inadvertently entered the central region of the host kidneys during the injection. They differentiated into nephrons, but did not contribute to chimera formation as they self-aggregated. Within the medulla, there are loops of Henle, distal tubules, and collecting ducts (Agarwal. *Am J Physiol Renal Physiol.* 2021; 321:F715-F739.). We have confirmed the presence of chimeric distal tubules that are ECAD positive (Fig. 2g, h. Prozialeck. *BMC Physiol.* 2004; 4: 10.). As NPCs differentiate into nephron structures ranging from glomeruli to distal tubules and extend in a convoluted manner towards the deeper regions of the kidney, nephrons originating later from the superficial CM might also reach the medulla. Consequently, distinguishing between nephrons originating at different times can be challenging.

Additionally, we detected a minority of chimeric collecting ducts (CK8+) originating from the UB (Supplementary Fig. 1a). As noted in the main text, we attribute this finding to the heterogeneous cell population of RPCs derived from mouse fetal kidneys used in this study.

4) The injection of human pluripotent stem cell derived NPC patch into host mouse does not present strong evidence: 1) It seems the injected hNPC patch formed their own structure instead of integrating to the host kidney. It's not chimeric kidney; 2) there is a lack result showing the maturation level of hNPC-derived nephrons; 3) why the hNPC-derived PTs did not express LTL?

Response:

1. As noted by the reviewer and acknowledged in our Discussion section, "their integration with host mouse nephrons, crucial for long-term evaluation, has yet to be established." "Nonetheless, even without a connection to the host, these *in vivo* human nephrons may still serve a valuable model for drug screening and pathological analysis, particularly within a limited timeframe."
2. We performed CD31 staining, which indicated that host vessels are supplying human nephrons (Fig. 8e). Transporter staining was also conducted (Fig. 8f), from which we cannot conclusively assert that they have matured sufficiently. Additionally, human nephron formation rate was lower compared to mouse RPCs (32 vs. 100%). Nevertheless, replicating cisplatin-induced nephropathy is still a significant achievement in our view. We believe that further efforts to generate long-term engrafted human chimeric nephrons and detailed assessments of their maturity are essential.
3. LTL was expressed in human proximal tubules, although to a lesser extent than in the host mouse. We have highlighted human proximal tubules with yellow arrows in Fig. 8g.

5) The application of the NNI method is limited. Firstly, the chimeric efficiency can be too low to demonstrate the phenotype in future application. Secondly, based on the current results present by the authors, it is very challenging to make mouse-human chimeric kidney. It may be helpful to discuss the possibility of using NNI method to study/validate genes that are associated with congenital kidney diseases.

Response:

As you pointed out, the chimeric efficiency was low and blood and urine tests were not conducted. However, this study demonstrated the feasibility of assessing a small number of donor nephrons through scRNA-seq, as mentioned in the Discussion section.

Regarding potential future applications, we have added the following to the Discussion section:

“This approach holds promise for studying congenital renal diseases such as Alport syndrome, polycystic kidney disease, nephronophthisis, as well as drug-induced nephrotoxicity, by modifying single causative genes or multiple renal tubular transporters.”

Reviewer #3

1) In Figure 2a, it appears that the GFP+ve donor cells are found only on one side of the kidney, likely indicating their presence primarily at the injection site. Similarly, in Figure 7d, which represents a 2-month-old chimeric kidney, there are very few localized GFP positive cells visible. The limited distribution of GFP+ve cells in both figures raises questions about the uniformity of the chimeric kidneys generated through RPC injection. It is possible that the injected RPCs did not populate the entire kidney uniformly. Please clarify.

Response:

Thank you for your comment regarding the distribution of donor cells. We have measured distribution of the donor cells in the cortical regions of the host kidney (Fig. 2d and Table 2). Donor cells were primarily localized to the injection site. It was previously noted that progenitor cells injected into the nephrogenic zone of fetal kidneys did not expand, and innovative approaches to enable a broader distribution of cell injection was explored (Takamura. *J. Clin. Med.* 2022, 11, 7237.).

2) In Sup Table 1, number of cells expressing GFP is reducing to 0.7% in 2 months (Cis-ve, normal condition) from the initial 1% suggests a decline in the population of GFP+ve cells over time under normal conditions. However, during cisplatin administration, there is an increase in the number of cells expressing GFP in 2 months. This indicates that cisplatin treatment might have influenced the survival or proliferation of cells (including GFP+ve cells), leading to an increase in their numbers compared to the normal condition. In Sup Figure 2c, Cis-ve (normal condition) 2-month group, fibroblast appears to be the major cell type expressing GFP. In Cis+

(cisplatin-treated) 2-month group, cell types like epithelial cells, macrophages, Intercalating cells, and Collecting Duct Principal cells, are the major cell types expressing GFP. Can you correlate these observations and suggest if the same kind of cellular profile is observed during kidney regeneration or any other explanation?

Response:

In each of the four groups, we conducted scRNA-seq on the trimmed regions including the GFP (+) areas, not the whole kidneys. Therefore, it is challenging to derive pathological significance from differences in the proportion of GFP (+) cells between these groups.

It also seems challenging to discuss the difference in GFP (+) cell ratios between the 2 months, CIS (-) group and the 2 months, CIS (+) group, due to the limited number of GFP (+) cells within each group, coupled with the fact that there is only one sample for each group. For instance, in the fibroblasts from the 2 months, CIS (-) group, the GFP (+) cell percentage appears high at 7.7%. However, the actual number of GFP (+) cells is only 3, so measurement errors can significantly impact the data.

Supplementary Table 1 and Supplementary Fig. 2c have been removed since discussing differences in GFP (+) cell ratios for each cell type between groups would deviate from the focus of this study, which is primarily on proximal tubules. In addition, the information presented in Supplementary Table 1 has been already mentioned in the main text and legends of Fig. 3, 4, and 6.

The following statement has been added as a limitation to the Discussion section: "Lastly, the assessment of drug-induced injury through scRNA-seq and immunostaining has primarily concentrated on proximal tubules. Further research is necessary to assess the pathophysiology of glomeruli and other tubular segments."

3) Line 169: "Their tubules got dilated after 1 month and disrupted after 2 months (Fig. 5a, b)". This is not evident from the figure.

4) Figure 5b: DAPI staining is not visible

Response:

We have replaced Fig. 5b with images that clearly show well-stained DAPI and the severe expansion of LTL-positive proximal tubules of the transplanted spheroids.

5) Line 177: "long-term engraftment (Fig. 5g)". The correct figure number is Fig 5h.

Response:

Thank you for your attention. We have updated the citation accordingly.

6) Figure 2d: What does the images in the bottom row represent? Please provide these details in the figure legend.

Response:

We have replaced the fluorescence stereomicroscopic images and modified the legend.

7) Line 189: Figure 6f not refereed in the manuscript.

Response:

Thank you for your attention. We have updated the citation accordingly.

REVIEWERS' COMMENTS:

Reviewer #1 (Remarks to the Author):

The authors have carefully addressed all the critics. I have no further questions. I think the revised manuscript reach the standard for publication in Communications Biology.

Reviewer #3 (Remarks to the Author):

The authors have addressed all raised queries and discussed the limitations of their study comprehensively. One technical limitation highlighted in this research is the limited chimerism ratio of transplanted renal progenitor cells (RPC). However, it's worth noting that the authors have demonstrated the maturation of these transplanted cells within neonatal kidneys and their response to drugs. This work has promise for researchers in organoids and artificial organs, serving as a valuable reference for planning and advancing their own research.